# Knowledge, perceptions, and use of psychedelics for mental health among autistic adults: An online survey

Sahba Afsharnia[1,2], Vivian Liang[1,2], Yona Lunsky[1,3], Aaron P. Orsini[4], Ami Tint[5], Hsiang-Yuan Lin[1,2,3]*

**1** Azrieli Adult Neurodevelopmental Centre, Campbell Family Mental Health Research Institute, Centre for Addiction and Mental Health, Toronto, Ontario, Canada, **2** Institute of Medical Science, Temerty Faculty of Medicine, University of Toronto, Toronto, Ontario, Canada, **3** Department of Psychiatry, Temerty Faculty of Medicine, University of Toronto, Toronto, Ontario, Canada, **4** Independent Researcher, San Francisco, California, United States of America, **5** Department of Psychology, Faculty of Arts, University of Calgary, Calgary, Alberta, Canada

* Hsiang-Yuan.Lin@camh.ca

## Abstract

Psychedelics such as psilocybin, LSD, and MDMA have shown promise in treating mental health conditions (e.g., depression, post-traumatic stress disorder) among neurotypical individuals, i.e., typically developing individuals without a diagnosed neurodevelopmental condition. However, their therapeutic potential for treating co-occurring mental-health conditions in autistic individuals remains under-explored. Autistic individuals often face co-occurring mental health challenges but are frequently excluded from clinical trials, creating a gap in effective treatments. This study aimed to explore knowledge, perceptions, and experiences of autistic adults regarding psychedelics. In this survey, "psychedelics" included classical psychedelics such as psilocybin and LSD, as well as MDMA. A cross-sectional online survey was conducted with English-speaking autistic adults. We assessed participants' knowledge of psychedelics, willingness to use them for mental health treatment, and any past psychedelic experiences. Data were analyzed using descriptive statistics and chi-square tests to assess group differences. A total of 424 participants began the survey, with 261 completing it. Nearly half resided in Canada. Participants generally viewed psychedelics positively, with 77.8% expressing a willingness to try them, and 69.7% reported past use—most commonly psilocybin mushrooms. Higher doses and highly meaningful experiences correlated with longer-lasting mental health improvements. Barriers included legal concerns, health risks, and logistical challenges. Participants with prior experience reported greater perceived knowledge and lower perceived risks. Autistic adults in this self-selecting sample demonstrated strong interest in psychedelics as potential treatments for mental health, despite significant barriers to access and research participation. These results highlight the importance

**Data availability statement:** The data relevant to this study are available from OSF at https://osf.io/zceyk/.

**Funding:** This study was supported by the Labatt Family Innovation Fund in Brain Health, Department of Psychiatry, University of Toronto and Canadian Institutes of Health Research Postdoctoral Fellowship. H.-Y.L. is supported by the Azrieli Adult Neurodevelopmental Centre at CAMH and an Academic Scholar Award from the Department of Psychiatry, University of Toronto. The funders had no role in the design of the study, data collection, analysis, decision to publish, or preparation of the manuscript.

**Competing interests:** The authors have declared that no competing interests exist.

of considering education, policy reform, and inclusive research practices to ensure that autistic people have opportunities to explore psychedelic therapies. These findings should be interpreted cautiously, as the sample may not be representative of the broader autistic population. Future trials should optimize dosing and explore long-term benefits of psychedelics in this population.

## Introduction

Autistic individuals (i.e., people with autism spectrum condition, ASC) frequently experience co-occurring mental health concerns at higher rates than neurotypical (i.e., typically developing individuals without a diagnosed neurodevelopmental condition) peers, often resulting in increased emergency department visits and hospitalizations [1]. Approximately 75% of autistic individuals have at least one co-occurring mental health condition, adversely affecting their own and their families' quality of life [1–4]. Current pharmacological interventions for these co-occurring mental health concerns associated with ASC, particularly anxiety, depression, and post-traumatic stress disorder (PTSD) are limited [5–8]. Many autistic individuals have suboptimal responses to standard pharmacological treatments and experience higher rates of side effects [5,9–11]. Co-occurring mental health concerns in autism are also linked with high treatment resistance [12]. Nevertheless, autistic individuals are generally excluded from clinical trials of new treatments, creating a significant gap in knowledge about effective interventions for this population. This exclusion perpetuates treatment disparities, impacts well-being, and neglects the unique mental health needs of autistic individuals [13].

Psychedelics, including classical psychedelics (acting primarily as agonists on the serotonin-2A, 5-HT2A receptor) and 3,4-methylenedioxy methamphetamine (MDMA), are a class of drugs which induce a range of psychoactive effects including cognitive, affective, and perceptual changes [14,15]. These effects have been shown to facilitate profound psychological insights and emotional breakthroughs, which are hypothesized to underlie their therapeutic potential [16]. Historically, classical psychedelics such as lysergic acid diethylamide (LSD), psilocybin, and ayahuasca (a brew containing the classical psychedelic DMT) have been used in various cultural and religious rituals.

The exploration of psychedelics as a therapeutic tool has gained considerable traction in recent years, with studies highlighting their potential benefits for a range of mental health conditions [17–19]. For example, studies have demonstrated that psilocybin can significantly reduce symptoms of depression [20–26] and anxiety [24,25] in neurotypical adults, with effects lasting from four weeks to several months after a single administration. MDMA has also demonstrated significant efficacy in reducing PTSD symptoms and functional impairment in a recent phase 3 clinical trial, marking it as a promising treatment for moderate to severe PTSD in diverse populations [27]. Recent meta-analyses and systematic reviews have reinforced the safety and efficacy of psychedelics in treating mental health conditions in neurotypical populations

[28–30]. The application of psychedelics in treating mental health conditions within neurodivergent populations, especially autistic individuals, remains largely unexplored, raising questions about the generalizability of these promising findings reported in neurotypical populations [9].

Early clinical trials with small sample sizes conducted in the 1960s and 1970s explored the effects of LSD on core symptoms in autistic children but were marred by methodological flaws and ethical concerns [31]. These early trials reported minimal side effects in autistic children, suggesting a potentially favourable safety profile [9]. However, early reports also described adverse effects such as aggression, self-injury, and agitation [9]. To date, only one modern-era pilot trial has explored the effects of MDMA in autistic adults (N = 12; 8 received MDMA in the randomized, double-blind phase). This study demonstrated a promising reduction in social anxiety alongside acceptable safety and tolerability [32]. Nonetheless, idiosyncratic response to treatment and vulnerability to side effects in autistic people [13] warrant further extensive investigation of the real-world experience of using psychedelics.

The first and only published online survey study on psychedelics in autistic adults was conducted by Stroud et al. [33] with 233 participants, primarily in the UK, who all reported past psychedelic use. Participants noted improvements in mental health and social outcomes, including reductions in anxiety, depression, and social difficulties following a single experience with classical psychedelics. However, individual factors such as mindset and prior expectations are known to influence psychedelic experiences and therapeutic outcomes [34]. While this study highlights potential benefits, it lacks detailed data on acute adverse events and persisting negative effects, such as the prevalence and impact of "bad experiences," underscoring the need to further explore, from the perspective of the community itself, both risks and benefits in this population. Whether the results of this first survey are reliable across different autistic cohorts remains unknown.

This study aimed to explore autistic adults' knowledge, perceptions, and experiences regarding psychedelics in relation to co-occurring mental-health conditions, rather than autism-specific symptoms. This study aimed to fill this gap by surveying autistic adults, who were fluent in English and able to provide online self-report. The primary research questions guiding this study were: (1) What are the general perspectives on, and knowledge about, psychedelics among autistic adults? (2) What are their opinions on using psychedelics as a treatment for co-occurring mental health symptoms? (3) Have they had any past experiences with psychedelics, and if so, what were their intentions and outcomes?

## Materials and methods

### Ethics statement

This study was approved by the Research Ethics Board at the Centre for Addiction and Mental Health (CAMH; REB #110/2022). Before participating in this anonymous survey online, all participants provided written informed consent via checkbox confirmation on the online platform. To ensure confidentiality, survey responses were stored on secure servers and only accessible to the research team. Participants were informed of the potential risks, including emotional distress from recalling past experiences with psychedelics, and were provided with resources for support.

### Participants

Inclusion criteria included participants who were 1) individuals aged 18 years and older, 2) self-identified or professionally diagnosed with ASC, 3) fluent in English, and 4) able to provide online self-report. Exclusion criteria were those who were unable to provide informed consent. Participants were recruited across Canada and internationally through multiple channels, including: (1) social media platforms (e.g., Twitter/X, Facebook, Instagram, Reddit); (2) autism and neurodevelopmental organizations such as the Health Care Access Research and Developmental Disabilities (H-CARDD) program, Autism Ontario, Autism Nova Scotia, AIDE Canada, and CASDA; (3) email invitations to previous CAMH research participants who had consented to be re-contacted; and (4) psychedelic and mental-health community networks such as the Autistic Psychedelic Community, Toronto Psychedelic Community, and the Psychedelic Association of Canada.

Recruitment materials were disseminated via posts, newsletters, and organizational mailing lists. Interested individuals could access the study link directly or request it via email. The recruitment period for this study began on March 10, 2023, and ended on March 31, 2024.

### Survey instrument

Study data were collected and managed using REDCap electronic data capture tools hosted at CAMH [35,36]. RED-Cap (Research Electronic Data Capture) is a secure, web-based software platform designed to support data capture for research studies, providing 1) an intuitive interface for validated data capture; 2) audit trails for tracking data manipulation and export procedures; 3) automated export procedures for seamless data downloads to common statistical packages; and 4) procedures for data integration and interoperability with external sources. The full survey questions are detailed in Supporting Information (S1 Text). The survey instrument consisted of three sections:

1. Demographics and Eligibility: This section (items 1–16) collected basic demographic information (e.g., age, sex assigned at birth, gender identity, educational background, mental health history) and confirmed eligibility criteria.

2. Interest in Knowledge and Perceptions of Psychedelics: This section (items 17–27) assessed participants' knowledge of psychedelics, perceptions of their safety and efficacy, and willingness to use them as a potential treatment for mental health conditions. Questions were adapted from existing surveys on psychedelics [37,38] and responses were recorded using Likert scales.

3. Experience with Psychedelics: This section (items 28–52) was completed only by participants who indicated prior use of psychedelics. It collected detailed information on the type and frequency of psychedelic use, the context of use, intentions for use, and perceived effects on mental health. To enhance participant comprehension and completion rates, dosage information (Section 3) was collected using simplified categorical ranges. Questions were adapted from the survey study by [39].

Embracing a co-production approach aligned with lived experience principles, we reviewed the survey questionnaire draft with autistic advisors at Azrieli Adult Neurodevelopmental Centre at CAMH to make this survey instrument more accessible to autistic populations. The survey items were accordingly modified to simplify sentence structure, substitute confusing terms, and add some explanations to enhance precision [40].

### Data collection and authenticity of responses

Participants received the survey link through various recruitment channels and provided informed consent before proceeding. The survey was anonymous, with no personally identifying information collected. Participants who completed the survey were invited to enter an optional raffle for a chance to win one of twelve Canadian Dollar $25 e-gift cards. Winners were randomly selected and contacted to claim their honorarium.

To ensure the integrity and authenticity of the responses, we implemented several measures to identify and remove fake or duplicate entries. To minimize the risk of duplicate or fraudulent responses, each email address could only be used once, and the first three characters of participants' postal codes were collected for validation. We manually checked all open-ended responses to ensure they were coherent and did not contain duplicated answers. Additionally, we reviewed the completed time of all survey submissions to confirm they were legitimate, with no instances of responses flooding in at unusual times (e.g., in the middle of the night). One participant did submit their response twice, and we included only the first submission.

Additionally, responses from individuals who skipped more than seven questions were excluded from the final dataset (although this exclusion did not apply to the third part on participants' experiences with psychedelics), in order to prevent multiple entries for the raffle and to ensure that only complete and meaningful data were included in the final analysis.

## Data analysis

Data were analyzed using descriptive statistics to summarize demographic information, knowledge, perceptions, and experiences with psychedelics. We also presented data based on the stratification of diagnosis (self-diagnosed vs. clinically diagnosed), gender (male, female, non-binary, or other), education (primary/secondary vs. post-secondary), age (younger vs. older (cutoff 40 years), marital status (married/partnered vs. single), and country of residence (Canadian vs. Non-Canadian). Secondary analyses used a chi-square test to assess between-group differences (e.g., comparing participants who had used psychedelics to those who had not, in addition to comparing against other conditions) in knowledge and experience. The Likert scale response of each item was classified into two or three classes. For example, for the three-class condition, meaningfulness was categorized as strong ("strongly meaningful," "extremely meaningful"), moderate ("moderately meaningful"), and low ("not meaningful at all," "slightly meaningful," "mildly meaningful"), and for the two-class condition, significance was grouped into high ("top 5 most significant," "very much," "single most significant") and low-moderate ("not at all," "slightly," "moderate"). Given the exploratory nature of this study, p-values for secondary analyses were not corrected for multiple comparisons; however, this increases the risk of Type I error, and these results should be interpreted as hypothesis-generating. For significant chi-square results, effect sizes (Phi or Cramer's V) were calculated and reported.

## Results

### Participant demographics

A total of 424 participants began the survey, with 261 completing it. The average age of the respondents was 33.4 years (SD = 12.8). All participants included in the final analyses self-identified or were professionally diagnosed as autistic, consistent with the study's inclusion criteria. The sample was predominantly assigned female at birth (62.5%). Regarding gender identity, 43.7% identified as women, 37.5% as men, 18.4% as non-binary, and 25.3% as other. For this question (and other questions throughout the survey) where the percentages do not add up to 100, the participants were allowed to select multiple answers. The majority of participants were white (87.4%) and resided mainly in Canada (45.6%) and the USA (42.1%). The sample was highly educated, with over two-thirds holding a college or graduate degree (67.8%). Half of the participants were single (50.6%). A significant portion of the sample reported having a history of co-occurring mental health conditions (96.6%), with anxiety disorders (85.1%) and mood disorders (67.0%) being the most common (Table 1).

### Interest in, knowledge, and perceptions of psychedelics

The majority of participants were aware that psychedelics are being researched for therapeutic potential (91.4%). When asked about their knowledge of psychedelics, most reported low knowledge (40.1%) or moderate knowledge (32.7%). Interest in learning about psychedelics was high, with the majority (76.7%) expressing strong interest. Participants' perceptions of the helpfulness of psychedelics were also positive: 69.7% believed they were very or extremely helpful. A significant majority, 77.8%, expressed willingness to try psychedelics in the future. When asked about their likelihood to participate in a government trial, most participants (72.7%) indicated a high likelihood. The primary barriers to trying psychedelics were work or transportation-related limitations (28.0%), legal status (23.3%), and health risks (18.3%). Interestingly, 33.1% of participants indicated that none of the listed reasons would prevent them from participating in a trial (Table 2).

### Previous experience with psychedelics

**General experience.** As shown in Table 3, a total of 181 participants (69.7%) reported using psychedelics at least once. Psilocybin mushrooms were the most commonly used psychedelic (91.7%), followed by LSD (55.2%) and MDMA (54.1%). The mean age of first psychedelic use was 24.7 years (SD = 10.6). Majority of participants (60.5%) used a mix of

**Table 1. Demographics and features.**

| Age | |
|---|---|
| <40 years old | 62.8% (164/261) |
| 40 years old and older | 37.2% (97/261) |
| **Sex** | |
| Female | 62.5% (163/261) |
| Male | 36.0% (94/261) |
| Intersex | 0.0% (0/261) |
| Prefer not to answer | 1.5% (4/261) |
| **Gender Identity[1]** | |
| Male | 37.5% (98/261) |
| Female | 43.7% (114/261) |
| Non-binary | 18.4% (48/261) |
| Other | 25.3% (66/261) |
| **Marital Status** | |
| Single[2] | 50.6% (132/261) |
| Married/In a committed relationship | 49.4% (129/261) |
| **Education** | |
| Did not complete high school/GED | 1.1% (3/261) |
| High school/GED | 9.2% (24/261) |
| Some college | 21.5% (56/261) |
| College graduate | 35.6% (93/261) |
| Some grad school or graduate | 32.2% (84/261) |
| **History of mental health conditions[3]** | |
| Any mental health disorder | 96.6% (252/261) |
| Anxiety disorder | 85.1% (222/261) |
| Eating disorder | 25.3% (66/261) |
| Impulse control disorder | 7.7% (20/261) |
| Mood disorder | 67.0% (180/261) |
| PTSD | 44.4% (116/261) |
| Obsessive Compulsive Disorder | 22.2% (58/261) |
| Neurodevelopmental disorder | 22.2% (58/261) |
| Personality disorder | 10.7% (28/261) |
| Psychotic disorder | 1.5% (4/261) |
| Substance use disorder | 25.7% (67/261) |
| Inertia, Meltdown, Shutdown | 48.7% (127/261) |
| Existential Crisis | 33.7% (88/261) |
| Other | 7.7% (20/261) |
| **Country of Residence** | |
| Canada | 45.6% (119/261) |
| USA | 42.1% (110/261) |
| Other[4] | 12.3% (32/261) |
| **Ethnicity** | |
| White | 87.4% (228/261) |
| Non-White[5] | 12.6% (33/261) |
| **Regular Drug Usage[3]** | |
| Alcohol | 21.8% (57/261) |
| Bath salt drug products | 0.4% (1/261) |

*(Continued)*

**Table 1.** (Continued)

| Age | |
|---|---|
| Benzodiazepines | 7.3% (19/261) |
| Caffeine | 69.7% (182/261) |
| Cannabis | 40.2% (105/261) |
| Dextromethorphan | 0.4% (1/261) |
| Hallucinogens | 9.2% (24/261) |
| Hash oil, dabs, THC oil, cannabis extract oils, wax, etc. | 12.3% (32/261) |
| Ketamine | 1.9% (5/261) |
| MDMA | 0.8% (2/261) |
| Methamphetamine | 0.0% (0/261) |
| Opioids | 2.7% (7/261) |
| Prescription antidepressants | 31.4% (82/261) |
| Prescription stimulants | 21.5% (56/261) |
| Synthetic marijuana | 0.4% (1/261) |
| Tobacco (nicotine) | 14.6% (38/261) |
| Other | 9.2% (24/261) |

[1]Gender identity was a multi-select question. Many participants chose more than one gender identity.

[2]"Single" includes participants who selected single, divorced, separated, and widowed as their marital status.

[3]This was a multi-select question.

[4]Other countries include Australia, Denmark, Finland, France, Germany, Ireland, Israel, Netherlands, Saudi Arabia, Sri Lanka, Sweden, Switzerland, and the UK.

[5]Non-White ethnicity includes South Asian (e.g., East Indian, Pakistani, Sri Lankan, etc.), Chinese, Black, Filipino, Latin American, Arab, Southeast Asian (e.g., Vietnamese, Cambodian, Laotian, Thai, etc.), West Asian (e.g., Iranian, Afghan, etc.), Korean, Japanese, First Nation, Unknown or not reported.

small to full doses, while 20.9% used full dose only. The primary intentions for psychedelic use were balanced between recreation (57.5%), psychological self-exploration (56.1%), and mental health support (57.5%). A majority of participants (68.6%) reported that their past use of psychedelics helped with mental health concerns.

**Psychedelic experience and mental health improvements.** As shown in Table 4, of the 147 participants who reported mental health improvements following psychedelic use, the most frequently reported benefits were for anxiety disorders (76.9%) and mood disorders (56.5%). Improvements were also noted for PTSD (40.1%), neurodevelopmental conditions such as ADHD (33.3%), and existential crises (34.7%). In terms of the duration of mental health improvements, nearly one-third (32.2%) reported benefits lasting 4 months or longer. Notably, 32.2% of participants described their symptoms as "greatly reduced" following the psychedelic experience. The substances associated with the most impactful experiences included psilocybin mushrooms (55.5%) and LSD (15.1%). Participants commonly cited moderate doses (32.9%) and mixed dosing strategies (11.6%) as contributing to mental health improvements. The setting of these experiences was primarily at home (63.9%) or in nature (33.3%).

Regarding the psychological impact of these experiences, majority of participants (91.%) rated their reference psychedelic experience as moderately or highly meaningful. Additionally, 77.3% of participants reported that the experience was moderately or highly psychologically insightful. The spiritual significance of these experiences was also notable, with 68.9% of participants rating their experience as moderately or highly significant in a spiritual context. When asked about how psychologically challenging this experience was, responses varied, with 43.4% reporting moderate challenge and 19.3% reporting high challenge (Table 5).

**Table 2. Knowledge and perception.**

| | Whole Sample (N = 257)[1] | Self-diagnosed (n = 51) | Medically Diagnosed (n = 206) |
|---|---|---|---|
| **Perceived Knowledge of Psychedelics** | | | |
| Low ("I don't know" or "slightly") | 40.1% (103/257) | 39.2% (20/51) | 40.3% (83/206) |
| Moderate | 32.7% (84/257) | 23.5% (12/51) | 35.0% (72/206) |
| High ("very" or "extremely") | 27.2% (70/257) | 37.3% (19/51) | 24.8% (51/206) |
| **Are you willing to try psychedelics in the future?** | | | |
| Yes | 77.8% (200/257) | 82.4% (42/51) | 76.7% (158/206) |
| Maybe | 14.4% (37/257) | 13.7% (7/51) | 14.6% (30/206) |
| No | 7.8% (20/257) | 3.9% (2/51) | 8.7% (18/206) |
| **Likeliness to Participate in a Government Trial[2]** | | | |
| Low ("not at all likely" or "slightly likely") | 14.8% (38/256) | 9.8% (5/51) | 16.0% (33/205) |
| Moderate | 10.2% (26/256) | 13.7% (7/51) | 9.2% (19/205) |
| High ("very likely" or "extremely likely") | 72.7% (186/256) | 76.5% (39/51) | 71.4% (147/205) |
| I don't know | 2.3% (6/256) | 0.0% (0/51) | 2.9% (6/205) |
| **Interest in Learning about Psychedelics** | | | |
| Low | 11.3% (29/257) | 9.8% (5/51) | 11.7% (24/206) |
| Moderate | 12.1% (31/257) | 7.8% (4/51) | 13.1% (27/206) |
| High | 76.7% (197/257) | 82.4% (42/51) | 75.2% (155/206) |
| **Perceived Helpfulness of Psychedelics** | | | |
| Low ("not helpful" or "somewhat") | 8.2% (21/257) | 5.9% (3/51) | 8.7% (18/206) |
| Moderate | 9.7% (25/257) | 7.8% (4/51) | 10.2% (21/206) |
| High ("very" or "extremely helpful") | 69.7% (179/257) | 78.4% (40/51) | 67.5% (139/206) |
| I don't know | 12.1% (31/257) | 7.8% (4/51) | 13.6% (28/206) |
| **Are Psychedelics Decriminalized Where You Live?** | | | |
| Yes | 15.2% (39/257) | 19.6% (10/51) | 14.1% (29/206) |
| No | 61.5% (158/257) | 58.8% (30/51) | 62.1% (128/206) |
| I don't know | 23.3% (60/257) | 21.6% (11/51) | 23.8% (49/206) |
| **To the best of your knowledge, are psychedelics being researched for therapeutic potential?** | | | |
| Yes | 91.4% (234/256) | 96.1% (49/51) | 90.2% (185/205) |
| No | 8.6% (22/256) | 3.9% (2/51) | 9.8% (20/205) |
| **What Might prevent you from trying Psychedelics?[3]** | | | |
| Legal status of psychedelics | 23.3% (60/257) | 21.6% (11/51) | 23.8% (49/206) |
| The health risks of psychedelics | 18.3% (47/257) | 15.7% (8/51) | 18.9% (39/206) |
| Past knowledge/experience with psychedelics | 10.9% (28/257) | 5.9% (3/51) | 12.1% (25/206) |
| Work or transportation-related limitations | 28.0% (72/257) | 21.6% (11/51) | 29.6% (61/206) |
| Religious beliefs | 1.6% (4/257) | 0.0% (0/51) | 1.9% (4/206) |
| Spiritual beliefs | 1.2% (3/257) | 2.0% (1/51) | 0.9% (2/206) |
| My mental health concerns are adequately managed with other treatments | 13.6% (35/257) | 7.8% (4/51) | 15.0% (31/206) |
| I don't think psychedelics would help with my concerns | 5.4% (14/257) | 5.9% (3/51) | 5.3% (11/206) |
| Other | 13.6% (35/257) | 15.7% (8/51) | 13.1% (27/206) |
| None of these reasons would prevent me from participating in a trial | 33.1% (85/257) | 45.1% (23/51) | 30.1% (62/206) |

[1]There are 4 participants who completely skipped this section of the survey.

[2]1 out of 257 participants did not respond to these items.

[3]This was a multi-select question.

**Table 3. General experiences of using psychedelics.**

| | Whole Sample (N = 181) | Self-diagnosed (N = 40) | Medically Diagnosed (N = 141) |
|---|---|---|---|
| **Mean age of first psychedelic experience** | 24.7 (SD = 10.6) | 23.1 (SD = 7.8) | 25.2 (SD = 11.3) |
| **Which substance(s) have you tried?[1]** | | | |
| Psilocybin | 91.7% (166/181) | 97.5% (39/40) | 90.1% (127/141) |
| LSD | 55.2% (100/181) | 67.5% (27/40) | 51.8% (73/141) |
| Morning glory seeds | 8.8% (16/181) | 10.0% (4/40) | 8.5% (12/141) |
| Mescaline | 8.8% (16/181) | 10.0% (4/40) | 8.5% (12/141) |
| Peyote Cactus | 5.5% (10/181) | 5.0% (2/40) | 5.7% (8/141) |
| San Pedro Cactus | 9.4% (17/181) | 10.0% (4/40) | 9.2% (13/141) |
| DMT | 19.3% (35/181) | 20.0% (8/40) | 19.1% (27/141) |
| Ayahuasca | 13.3% (24/181) | 17.5% (7/40) | 12.1% (17/141) |
| MDMA | 54.1% (98/181) | 65.0% (26/40) | 51.1% (72/141) |
| Percentage that has only tried 1 substance | 26.5% (48/181) | 17.5% (7/40) | 29.1% (41/141) |
| Percentage that has tried 2–3 substances | 19.3% (35/181) | 45.0% (18/40) | 45.4% (64/141) |
| Percentage that has tried 4 + substances | 26.0% (47/181) | 37.5% (15/40) | 22.7% (32/141) |
| **What sort of dosing do you typically use?[3]** | | | |
| Small doses/microdoses only | 18.6% (32/172) | 5.1% (2/39)[2] | 22.6% (30/133) |
| Mixed use of various doses, from small to full doses | 60.5% (104/172) | 74.4% (29/39)[2] | 56.4% (75/133) |
| Full doses only | 20.9% (36/172) | 20.5% (8/39)[2] | 21.1% (28/133) |
| **Intention for psychedelic experience[1]** | | | |
| No serious intention, other people were using | 10.0% (18/181) | 5.0% (2/40) | 11.3% (16/141) |
| Curiosity | 28.2% (51/181) | 22.5% (9/40) | 29.8% (42/141) |
| Recreation | 57.5% (104/181) | 57.5% (23/40) | 57.4% (81/141) |
| Psychological self-exploration | 56.1% (107/181) | 72.5% (29/40) | 55.3% (78/141) |
| Explore spirituality or the sacred | 37.6% (68/181) | 50.0% (20/40) | 34.0% (48/141) |
| To help with mental health concerns | 57.5% (104/181) | 57.5% (23/40) | 57.4% (81/141) |
| **Time since last experience** | | | |
| Within 24 hours | 8.8% (16/181) | 7.5% (3/40) | 9.2% (13/141) |
| Within the past week | 17.7% (32/181) | 25.0% (10/40) | 15.6% (22/141) |
| Within the past month | 14.4% (26/181) | 17.5% (7/40) | 13.5% (19/141) |
| Within the past year | 30.9% (56/181) | 22.5% (9/40) | 33.3% (47/141) |
| In the past 5 years | 12.7% (23/181) | 12.5% (5/40) | 9.2% (13/141) |
| 5-10 years | 5.0% (9/181) | 2.5% (1/40) | 5.7% (8/141) |
| More than 10 years | 7.7% (14/181) | 10.0% (4/40) | 7.1% (10/141) |
| **Did your past psychedelic use help with your mental health concerns?[3]** | | | |
| Yes | 68.6% (118/172) | 76.9% (30/39) | 66.2% (88/133) |
| No | 14.5% (25/172) | 10.3% (4/39) | 15.8% (21/133) |
| I don't know/Not sure | 11.0% (19/172) | 10.3% (4/39) | 11.3% (15/133) |
| Other | 5.8% (10/172) | 2.6% (1/39) | 6.8% (9/133) |

[1]This was a multi-select question.

[2]Significant Chi-square tests (uncorrected p < .05), suggesting significant effects of the identified demographic factors on results.

[3]172 out of 181 participants who had experiences using psychedelics responded to this item.

**Table 4. Psychedelic experience that led to improvements in your mental health.**

| | Whole Sample (N = 147) | Self-diagnosed (N = 35) | Medically Diagnosed (N = 112) |
|---|---|---|---|
| **Which of these mental health conditions improved after using psychedelics?[1]** | | | |
| Anxiety Disorder (Social Anxiety, Generalized Anxiety, Panic, Phobia, etc.) | 76.9% (113/147) | 85.7% (30/35) | 74.1% (83/112) |
| Eating Disorder (Anorexia, Bulimia, etc.) | 10.9% (16/147) | 14.3% (5/35) | 9.8% (11/112) |
| Impulse Control Disorder (Pyromania, Compulsive Gambling, etc.) | 4.1% (6/147) | 2.9% (1/35) | 4.5% (5/112) |
| Mood Disorder (Depression, Mania, Bipolar, etc.) | 56.5% (83/147) | 62.9% (22/35) | 54.5% (61/112) |
| Post-traumatic Stress Disorder (PTSD) | 40.1% (59/147) | 42.9% (15/35) | 39.3% (44/112) |
| Obsessive Compulsive Disorder (OCD) | 12.9% (19/147) | 11.4% (4/35) | 13.4% (15/112) |
| Personality Disorder (Paranoid, Avoidant, Borderline, Narcissistic, etc.) | 8.8% (13/147) | 17.1% (6/35) | 6.3% (7/112) |
| Psychotic Disorder (Schizophrenia, Schizoaffective, etc.) | 1.4% (2/147) | 2.9% (1/35) | 0.9% (1/112) |
| SubstanceRelated Disorder (Alcohol or Drug Dependence) | 17.0% (25/147) | 17.1% (6/35) | 17.0% (19/112) |
| Neurodevelopmental Disorders (Attention deficit/hyperactivity disorder, ADHD, Learning Disorders, Tics, etc.) | 33.3% (49/147) | 40.0% (14/35) | 31.3% (35/112) |
| Burnout, Inertia, Meltdown, and Shutdown | 26.5% (39/147) | 28.6% (10/35) | 25.9% (29/112) |
| Existential Crisis | 34.7% (51/147) | 31.4% (11/35) | 35.7% (40/112) |
| Other | 11.6% (17/147) | 11.4% (4/35) | 11.6% (13/112) |
| **Mean age at which the psychedelic experience that improved mental health took place** | 29.9 (SD = 11.2) | 28.8 (SD = 8.4) | 30.2 (SD = 12.0) |
| **How would you describe your mental health improvement after this psychedelic experience?** | | | |
| Stopped experiencing the mental health concerns completely since the experience (full remission). | 2.7% (4/146) | 0.0% (0/35) | 3.6% (4/111) |
| Greatly reduced experiencing the mental health concern(s) since the experience. | 32.2% (47/146) | 37.1% (13/35) | 30.6% (34/111) |
| Reduced experiencing the mental health concern(s) somewhat since the experience. | 24.7% (36/146) | 34.3% (12/35) | 21.6% (24/111) |
| Initially stopped experiencing the mental health concern(s) **completely**, then the mental health concern(s) returned to the same level as before. | 8.2% (12/146) | 2.9% (1/35) | 9.9% (11/111) |
| Stopped experiencing the mental health concern(s) somewhat for a period of time, then the mental health concern(s) returned at the same level as before. | 16.4% (24/146) | 14.3% (5/35) | 17.1% (19/111) |
| Other | 15.8% (23/146) | 11.4% (4/35) | 17.1% (19/111) |
| **How long did your mental health improvement last?** | | | |
| Less than 1 week | 9.6% (14/146) | 8.6% (3/35) | 9.9% (11/111) |
| 1 - 2 weeks | 13.0% (19/146) | 14.3% (5/35) | 12.6% (14/111) |
| 3 - 4 weeks | 6.8% (10/146) | 14.3% (5/35) | 4.5% (5/111) |
| 1 - 3 months | 8.9% (13/146) | 8.6% (3/35) | 9.0% (10/111) |
| 4 - 6 months | 15.1%(22/146) | 11.4% (4/35) | 16.2% (18/111) |
| 7 - 12 months | 8.2% (12/146) | 0.0% (0/35) | 10.8% (12/111) |
| 1 - 2 years | 17.1% (25/146) | 14.3% (5/35) | 18.0% (20/111) |
| 3 - 5 years | 9.6% (14/146) | 11.4% (4/35) | 9.0% (10/111) |
| 6 - 10 years | 3.4% (5/146) | 2.9% (1/35) | 3.6% (4/111) |
| 11 - 20 years | 4.1% (6/146) | 8.6% (3/35) | 2.7% (3/111) |
| More than 20 years | 4.1% (6/146) | 5.7% (2/35) | 3.6% (4/111) |
| **Which substance led to the psychedelic experience associated with your improvement in mental health concerns?** | | | |
| Psilocybin mushrooms | 55.5% (81/146) | 42.9% (15/35) | 59.5% (66/111) |
| LSD | 15.1% (22/146) | 25.7% (9/35) | 11.7% (13/111) |
| Morning glory seeds | 0.0% (0/146) | 0.0% (0/35) | 0.0% (0/111) |
| Mescaline (pure compound) | 0.0% (0/146) | 0.0% (0/35) | 0.0% (0/111) |
| Peyote cactus | 1.4% (2/146) | 0.0% (0/35) | 1.8% (2/111) |

*(Continued)*

**Table 4.** (Continued)

| | Whole Sample (N = 147) | Self-diagnosed (N = 35) | Medically Diagnosed (N = 112) |
|---|---|---|---|
| San Pedro cactus | 2.1% (3/146) | 0.0% (0/35) | 2.7% (3/111) |
| DMT (pure compound) | 2.7% (4/146) | 2.9% (1/35) | 2.7% (3/111) |
| Ayahuasca | 0.7% (1/146) | 2.9% (1/35) | 0.0% (0/111) |
| MDMA (ecstasy, Molly) | 13.0% (19/146) | 22.9% (8/35) | 9.9% (11/111) |
| Other | 9.6% (14/146) | 2.9% (1/35) | 11.7% (13/111) |
| **Psychedelic doses improving mental health issues in those with positive experiences[2]** | | | |
| Very low | 9.6% (14/146) | 8.6% (3/35) | 9.9% (11/111) |
| Low | 10.3% (15/146) | 14.3% (5/35) | 9.0% (10/111) |
| Moderate | 32.9% (48/146) | 34.3% (12/35) | 32.4% (36/111) |
| High | 13.7% (20/146) | 17.1% (6/35) | 12.6% (14/111) |
| Very high | 8.9% (13/146) | 5.7% (2/35) | 9.9% (11/111) |
| Mixed doses | 11.6% (17/146) | 8.6% (3/35) | 12.6% (14/111) |
| **Location of psychedelic experience that helped with mental health concerns[1]** | | | |
| Home | 63.9% (94/147) | 60.0% (21/35) | 66.1% (74/112) |
| Party | 5.4% (8/147) | 2.8% (1/35) | 6.3% (7/112) |
| Public place | 3.4% (5/147) | 0.0% (0/35) | 4.5% (5/112) |
| Concert | 9.5% (14/147) | 14.3% (5/35) | 8.0% (9/112) |
| Nature | 33.3% (49/147) | 35.3% (12/35) | 33.0% (37/112) |
| Religious | 7.5% (11/147) | 11.4% (4/35) | 6.3% (7/112) |
| Other | 8.8% (13/147) | 8.6% (3/35) | 8.9% (10/112) |

[1]This was a multi-select question.

[2]1 out of 147 participants did not respond to this item.

Behavioral changes following the psychedelic experience were widespread. The most frequently reported change was improved relationships, as noted by 63.4% of participants. Other notable changes included an increase in exercise (31.0%), reduced or quitting other drugs (30.3%), and career improvements (30.3%). Negative behavioural effects were minimal, with 5.5% reporting worsened relationships and 4.8% noting worsened careers. Participants also reported various aspects of how their psychedelic experience contributed to their mental health improvement: Key factors included strengthening their belief in their ability to recover (67.6%), reducing the stress involved with recovering (62.1%), and changes in life priorities or values (60.7%) (Table 5).

Of 170 participants who had tried psychedelics, 81.8% reported no persisting negative effects, 10.0% (n = 17) reported some, and 8.2% (n = 14) were unsure. Overall, 87.5% rated any negative effects as not at all or slightly severe, 8.9% as moderately severe, and 3.6% as very or extremely severe (Table 5). Based on Item 50, "Please describe any negative or potentially negative persisting effects you may have experienced," among the 17 participants who initially endorsed persisting negative effects in the closed-ended question, four did not describe substantive psychedelic-related issues in their open-ended response (e.g., "initially depressed but became better"). The remaining 13 indicated these negative effects lasted for a few days to a few weeks, including short-term anxiety or panic (n = 7), heightened sensory sensitivities (n = 3), cravings leading to more frequent use (n = 2), longer comedowns (n = 2), physical strain (n = 1), perceptual distortions (n = 2), or ego dissolution (n = 1). One participant reported experiencing multiple adverse effects—mirroring those described by others—after consuming a dose approximately twice their usual therapeutic level. Two participants attributed these challenges to confronting unsatisfying life or relationship situations and noted eventual relief through counseling or reduced use.

**Table 5. Feelings, thoughts, and experiences during the overall psychedelic experience that led to improvements in your mental health.**

| | Whole Sample (N=146)[1] | Self-diagnosed (n=35) | Medically Diagnosed (n=111) |
|---|---|---|---|
| **How personally meaningful was the psychedelic experience that led to mental health improvement?** | | | |
| Low meaningfulness ("not meaningful at all" or "slightly meaningful") | 8.9% (13/146) | 8.6% (3/35) | 9.0% (10/111) |
| Moderate meaningfulness ("mildly meaningful" or "moderately meaningful") | 38.4% (56/146) | 40.0% (14/35) | 37.8% (42/111) |
| High meaningfulness ("strongly meaningful" or "extremely meaningful") | 52.7% (77/146) | 51.4% (18/35) | 53.2% (59/111) |
| **Degree to which the psychedelic experience that led to your improvement in mental health issues were spiritually significant to you.[2]** | | | |
| Low significance ("not at all" or "slightly") | 31.0% (45/145) | 20.6% (7/34) | 34.2% (38/111) |
| Moderate significance ("moderately" or "very much") | 37.9% (55/145) | 50.0% (17/34) | 34.2% (38/111) |
| High significance ("among the 5 most spiritually significant experiences of my life" or "the single most spiritually significant experience of my life") | 31.0% (45/145) | 29.4% (10/34) | 22.5% (35/111) |
| **How psychologically challenging was the psychedelic experience that led to improvements in your mental health concerns?[2]** | | | |
| Low challenge ("not challenging at all" or "slightly challenging") | 37.2% (54/145) | 35.3% (12/34) | 37.8% (42/111) |
| Moderate challenge ("mildly challenging" or "moderately challenging") | 43.4% (63/145) | 41.2% (14/34) | 44.1% (49/111) |
| High challenge ("strongly challenging" or "extremely challenging") | 19.3% (28/145) | 23.5% (8/34) | 18.0% (20/111) |
| **How psychologically insightful to you was the psychedelic experience that led to your improvement in mental health concerns?** | | | |
| Low insight ("not psychologically insightful at all" or slightly psychologically insightful") | 22.8% (33/145) | 14.7% (5/34) | 25.2% (28/111) |
| Moderate insight ("mildly insightful" or "moderately insightful") | 21.4% (31/145) | 23.5% (8/34) | 20.7% (23/111) |
| High insight ("strongly psychologically insightful" or extremely psychologically insightful") | 55.9% (81/145) | 61.8% (21/34) | 54.1% (60/111) |
| **Items related to your psychedelic associated improvement in mental health concerns.[2,5]** | | | |
| Strengthening your belief in your own ability to recover | 67.6% (98/145) | 70.6% (24/34) | 66.7% (74/111) |
| Reducing stress involved with recovering. | 62.1% (90/145) | 73.5% (25/34) | 58.6% (65/111) |
| Reframing mental health improvement as a spiritual task. | 46.2% (67/145) | 64.7% (22/34) | 40.5% (45/111) |
| Changing life priorities or values. | 60.7% (88/145) | 76.5% (26/34) | 55.9% (62/111) |
| Changing your orientation toward the future, so that longterm benefits outweighed immediate desires/preoccupations. | 46.9% (68/145) | 44.1% (15/34) | 47.7% (53/111) |
| **Other behavioral changes after psychedelic experience[2,5]** | | | |
| None | 30.3% (44/145) | 20.6% (7/34) | 33.3% (37/111) |
| Reduced/quit other drugs | 30.3% (44/145) | 32.4% (11/34) | 29.7% (33/111) |
| Started using other drugs | 6.2% (9/145) | 2.9% (1/34) | 7.2% (8/111) |
| Changes in diet | 24.1% (35/145) | 35.3% (12/34) | 20.7% (23/111) |
| Increased exercise | 31.0% (45/145) | 47.1% (16/34) | 26.1% (29/111) |
| Decreased exercise | 2.8% (4/145) | 2.9% (1/34) | 2.7% (3/111) |
| Improved relationships | 63.4% (92/145) | 79.4% (27/34) | 58.6% (65/111) |
| Worsened relationships | 5.5% (8/145) | 5.9% (2/34) | 5.4% (6/111) |
| Improved career | 30.3% (44/145) | 32.4% (11/34) | 29.7% (33/111) |
| Worsened career | 4.8% (7/145) | 2.9% (1/34) | 5.4% (6/111) |
| **Did you experience any persisting negative effects from this psychedelic experience?[3]** | | | |
| Yes | 10.0% (17/170) | 14.3% (4/38) | 9.8% (13/132) |
| No | 81.8% (139/170) | 86.8% (33/38) | 80.3% (106/132) |
| Not sure | 8.2% (14/170) | 2.6% (1/38) | 9.8% (13/132) |
| **Overall, how would you rate the severity of these negative effects?[4]** | | | |
| Low ("not at all severe" or "slightly severe") | 87.5% (147/168) | 86.8% (33/38) | 87.7% (114/130) |
| Moderate ("moderately severe") | 8.9% (15/168) | 10.5% (4/38) | 8.5% (11/130) |
| High ("very severe" or "extremely severe") | 3.6% (6/168) | 2.6% (1/38) | 3.8% (5/130) |

[1]Items in these tables were branched out questions, thus only 146 out of 181 participants who had experiences using psychedelics reported.

[2]1 out of 146 participants specified in the specified 1 did not respond to these items.

[3]170 out of 181 participants who had experiences using psychedelics responded to this item.

[4]2 out of 170 participants who reported negative experiences using psychedelics did not respond to this item.

[5]This was a multi-select question.

## Effects of demographics

We performed Chi-square tests ($p_{uncorrected} < 0.05$) to assess how demographic factors influenced participants' perspectives and experiences with psychedelics. As shown in S1 Table, perceived knowledge varied by both age (≥40 vs. <40; $\chi^2(2, N=257) = 7.52$, $p_{uncorrected} = 0.023$; $V = 0.171$) and country of residence (Canadian vs. non-Canadian; $\chi^2(2, N=257) = 21.28$, $p_{uncorrected} < 0.001$; $V = 0.288$). Canadians also differed from non-Canadians in their willingness to try psychedelics ($\chi^2(2, N=257) = 10.98$, $p_{uncorrected} = 0.004$; $V = 0.206$), interest in learning ($\chi^2(2, N=257) = 7.42$, $p_{uncorrected} = 0.024$; $V = 0.170$), perceived helpfulness ($\chi^2(3, N=257) = 24.81$, $p_{uncorrected} < 0.001$; $V = 0.310$), and awareness of decriminalization status ($\chi^2(2, N=257) = 32.14$, $p_{uncorrected} < 0.001$; $V = 0.353$). Moreover, Canadian participants reported fewer multiple-psychedelic experiences ($\chi^2(2, N=177) = 10.43$, $p_{uncorrected} = 0.005$; $V = 0.243$) and were less likely to indicate that psychedelics helped with their mental health ($\chi^2(3, N=172) = 9.77$, $p_{uncorrected} = 0.021$; $V = 0.239$), compared to non-Canadians (S2 Table). Diagnostic status (self- vs. clinically-diagnosed) was associated with the typical dose used ($\chi^2(2, N=172) = 6.61$, $p_{uncorrected} = 0.037$; $V = 0.195$), with self-diagnosed participants more likely to use mixed doses and fewer small doses (Table 3). Among those who found psychedelics beneficial, assigned sex at birth (female vs. male; $\chi^2(4, N=106) = 11.60$, $p_{uncorrected} = 0.021$; $V = 0.331$) and education level (high school vs. post-secondary; $\chi^2(4, N=113) = 10.15$, $p_{uncorrected} = 0.038$; $V = 0.300$) influenced which dose was perceived as most effective (S3 Table). Additionally, females reported greater psychological challenge ($\chi^2(2, N=141) = 8.78$, $p_{uncorrected} = 0.012$; $V = 0.249$) yet also higher insight ($\chi^2(2, N=141) = 6.51$, $p_{uncorrected} = 0.039$; $V = 0.216$) from their experiences, relative to males (S4 Table).

## Secondary analyses

Secondary analyses revealed how perceived knowledge, psychedelic type, and dosage influenced reported outcomes.

**Dosage.** Among participants who reported mental health improvements versus no improvements, there was no significant difference in the type of dose used ($\chi^2(2, N=172) = 0.32$, $p_{uncorrected} = 0.854$); both groups primarily used mixed doses. However, dosage was significantly associated with how long improvements lasted ($\chi^2(3, n=137) = 9.47$, $p_{uncorrected} = .024$; $V = 0.263$). Pairwise comparisons indicated that high doses produced longer-lasting benefits than either low ($\chi^2(1, n=62) = 5.65$, $p_{uncorrected} = 0.017$; $\Phi = 0.302$) or moderate doses ($\chi^2(1, n=91) = 7.99$, $p_{uncorrected} = 0.005$; $\Phi = 0.296$). In contrast, participants' dosage type (small, full, or mixed) was not related to reports of lasting negative effects ($\chi^2(2, n=170) = 3.00$, $p_{uncorrected} = 0.224$).

**Type of psychedelic.** No significant relationship emerged between the type of psychedelic (e.g., psilocybin, LSD) and the duration of mental health improvements ($\chi^2(3, n=146) = 7.62$, $p_{uncorrected} = 0.055$), perceived personal meaningfulness ($\chi^2(3, n=146) = 1.04$, $p_{uncorrected} = 0.791$), spiritual significance ($\chi^2(3, n=145) = 2.93$, $p_{uncorrected} = 0.403$), or lasting negative effects ($\chi^2(3, n=145) = 4.38$, $p_{uncorrected} = 0.223$).

**Perceived knowledge.** A significant relationship was found between perceived knowledge of psychedelics and participants' self-reported mental health improvements ($\chi^2(2, n=172) = 7.70$, $p_{uncorrected} = 0.021$; $V = 0.212$). Those reporting high knowledge were more likely to endorse positive outcomes than those reporting low knowledge ($\chi^2(1, n=109) = 7.54$, $p_{uncorrected} = 0.006$; $\Phi = 0.263$). No significant difference was observed between moderate and high knowledge ($\chi^2(1, n=130) = 1.01$, $p = 0.314$) or moderate vs. low knowledge ($\chi^2(1, n=105) = 3.29$, $p_{uncorrected} = 0.070$). Further, having prior psychedelic experience was associated with both higher perceived knowledge ($\chi^2(2, n=254) = 70.95$, $p_{uncorrected} < 0.001$; $V = 0.529$) and lower perceived danger ($\chi^2(2, n=254) = 18.69$, $p_{uncorrected} < 0.001$; $V = 0.271$), specifically when comparing participants who classified themselves as low vs. moderate or high knowledge/danger ($\chi^2(1, n=183) = 30.04$, $p_{uncorrected} < 0.001$; $\chi^2(1, n=170) = 58.03$, $p_{uncorrected} < .001$; $\chi^2(1, n=87) = 17.55$, $p_{uncorrected} < 0.001$).

**Impact of psychedelic experiences.** The meaningfulness of participants' most impactful psychedelic experience was also significantly associated with how long mental health improvements lasted ($\chi^2(2, n=146) = 12.90$, $p_{uncorrected} = 0.002$; $V = 0.297$). Pairwise comparisons showed that the high-meaningfulness group reported longer-lasting benefits compared to the moderate-meaningfulness groups ($\chi^2(1, n=121) = 8.34$, $p_{uncorrected} = 0.004$; $\Phi = 0.263$) and the low-meaningfulness

group ($\chi^2$(1, n = 102) = 7.81, $p_{uncorrected}$ = 0.005; $\Phi$ = 0.277). In contrast, there was no significant difference between the low- and moderate-meaningfulness groups ($\chi^2$(1, n = 69) = 0.224, $p_{uncorrected}$ = 0.636). Lastly, participants with prior experience expressed a greater interest in learning about psychedelics than those without such experience ($\chi^2$(1, n = 254) = 48.17, $p_{uncorrected}$ < 0.001; $\Phi$ = 0.435).

## Discussion

This study offers a novel exploration of the perspectives of autistic adults on psychedelics, providing critical insights into the potential benefits and challenges of integrating these psychoactive substances into mental health treatments for this population. The findings reveal a strong interest in psychedelics as therapeutic tools, with the majority of autistic adult participants perceiving them as beneficial and expressing a willingness to participate in future research. This openness is particularly significant given the historical exclusion of autistic individuals from clinical trials, highlighting a crucial opportunity to bridge this research gap [13]. The secondary analyses revealed key patterns related to dosage and the meaningfulness of the psychedelic experience. Higher doses were linked to longer-lasting mental health improvements, while highly meaningful experiences were strongly related to more enduring benefits. Additionally, perceived knowledge and past experience with psychedelics significantly influence the perceived danger, interest in future research, and response to psychedelics.

Our findings can be benchmarked against several earlier published surveys. For example, we found that "recreation/ fun" and "self-exploration" were two of the primary motivations for psychedelic use, consistent with the Canadian Psychedelic Survey's findings [13]. However, our study also highlighted the use of psychedelics for "helping mental health concerns" as a key motivator among autistic adults, which is different from intentions for psychedelic uses in the general Canadian population [41]. Our findings regarding the willingness to learn more about psychedelics can be compared to the results of Glynos et al. who surveyed individuals with fibromyalgia [37]. Our participants, like theirs, expressed significant interest in psychedelics as a therapeutic option, with the majority perceiving them as beneficial. Notably, autistic participants in our study appear to have higher perceived knowledge of psychedelics than individuals with fibromyalgia [37]. This suggests that autistic adults may be more informed or motivated to learn about psychedelics, possibly due to higher rates of co-occurring mental health conditions.

In terms of public perceptions, while the neurotypical population may harbor legal and safety concerns [38], our participants showed more willingness to engage in research, suggesting that at least a subset of autistic adults may be more open to alternative mental health treatments. This greater openness could reflect characteristics of our self-selecting sample (such as previous psychedelic use or favorable attitudes toward psychedelics) or it could relate to unique perspectives within the autistic community. However, the current study design does not allow us to determine which of these factors, or both, contributed to this finding. Despite positive perceptions, significant barriers to psychedelic research participation were identified, including legal concerns, health risks, and logistical challenges such as transportation. These barriers highlight the need for systemic changes to improve access to psychedelic therapies. Addressing these concerns through policy reform and targeted education can help ensure that autistic individuals are not excluded from the potential benefits of psychedelic-assisted therapies [13].

Our findings further underscore the broad applicability of psychedelics for addressing diverse mental health challenges in autistic adults [9,13,19,33]. Anxiety was the most frequently reported area of improvement, affecting 76.9% of participants, which aligns with reports that psychedelics can alleviate anxiety in other populations [20,24–26]. Mood disorders and PTSD also showed significant gains, consistent with prior research demonstrating the robust therapeutic potential of psychedelic-assisted treatments in neurotypical samples [23,26,27,32]. Notably, many participants reported benefits lasting over half a year, reflecting the durability of one-time psychedelic-assisted treatments found in neurotypical populations [19,22,24–26,42]. Moderate doses (32.9%) were reported as most effective, followed by high doses (13.7%) and mixed doses (11.6%), reinforcing the importance of dosage in determining outcomes. The role of setting was pivotal, echoing the long-standing emphasis on "set and setting" in psychedelic research [19,32,42,43]. Psilocybin mushrooms emerged

as the most commonly used psychedelic in our sample, potentially reflecting perceived safety and therapeutic benefits [9,27,33,44]. Although these data highlight the promise of psychedelics as novel tools for addressing co-occurring psychiatric conditions in autistic individuals, future investigations should evaluate how different dosages, controlled settings, and integration strategies can optimize outcomes in this population [9,13,19,45]. In addition, protocols should account for the unique sensory and emotional profiles of autistic individuals to mitigate potential adverse effects and maximize benefits [9,13,32].

Most participants described their psychedelic experiences as meaningful, insightful, and beneficial. Individuals assigned female at birth reported more psychologically challenging experiences yet also gained greater insight than those assigned male at birth, aligning with prior evidence that sex may shape subjective response [46]. Among participants with psychedelic experience, 87.5% rated their experience as not at all or slightly negative, paralleling findings from a UK-based survey in autistic adults [33]. A small subgroup (7.6%, based on refined analysis) endorsed persisting negative effects—such as short-term anxiety, sensory sensitivities, or cravings—that lasted days to weeks, consistent with observations in neurotypical samples [47]. Notably, some of these individuals ultimately resolved their challenges through counseling, suggesting that negative effects can sometimes be mitigated with appropriate support. Although safety concerns are valid, our findings indicate that risks of long-term harm may be relatively low in relation to potential therapeutic benefits in this sample. However, only three items in our survey specifically addressed negative experiences, underscoring a need for more comprehensive assessments. Future research should incorporate structured monitoring and validated instruments to systematically capture the frequency, severity, and duration of potential negative outcomes. Given the heightened vulnerabilities of autistic individuals—such as sensory sensitivities and co-occurring mental health challenges—thorough screening, preparation, and follow-up integration are crucial. Balancing harm reduction with equitable access is especially important, as many are already experimenting with psychedelics independently. Future studies should prioritize co-design methodologies, actively involving autistic advisors and community members, to capture both beneficial and adverse aspects of psychedelic use, thereby guiding the development of safer and more effective interventions.

Our results also showed a relationship between prior psychedelic experience and higher perceived knowledge. Those with higher perceived knowledge were more likely to report positive mental health outcomes. Together, these suggest that direct experience and familiarity with psychedelics can enhance understanding, awareness, and perceptions of safety and efficacy, consistent with a prior study on neurotypical adults [44]. This highlights the importance of education and extensive informed consent before psychedelic-assisted therapy for autistic people.

We also examined differences between self-diagnosed (n = 51) and clinically diagnosed (n = 206) autistic adults. Overall, the two groups showed similar knowledge, perceptions, and outcomes related to psychedelic use (Tables 2, 4 and 5). The only statistically significant difference observed was in dosing practices, with self-diagnosed participants more likely to use mixed doses and less likely to use microdoses compared to clinically diagnosed participants (Table 3). While the inclusion of self-diagnosed individuals enhances the inclusivity of the sample—reflecting the reality that many autistic adults face barriers to formal diagnosis—it also introduces variability in diagnostic certainty. However, the general consistency in responses suggests that self-identified autistic adults in this sample share similar perspectives on psychedelics with their clinically diagnosed peers.

Our findings align with the first and only published survey on psychedelic use in autistic adults [33], which found a similar prevalence of use (~70%) and preference for classic psychedelics, particularly psilocybin. Both studies highlight potential therapeutic benefits, with participants reporting improvements in their mental health, psychological distress, anxiety, and well-being following psychedelic use. These positive experiences are similarly reported in adults with substance misuse [39] and in the Canadian Psychedelic Survey [41], suggesting that psychedelics can provide meaningful and lasting psychological benefits across diverse populations. However, consistent with the Canadian Psychedelic Survey [41] and Stroud et al. [33], our findings acknowledged the potential (in a small proportion of participants) for challenging experiences, underscoring the need for careful preparation and integration support. While acknowledging the limitations

of self-reported data, these converging results encourage further research into the clinical applications of psychedelics for autistic adults.

One of our key findings is the significant relationship between higher doses of psychedelics and longer-lasting mental health improvements, suggesting that optimal dosing strategies are essential for maximizing therapeutic outcomes in autistic adults. This dose-dependent relationship aligns with existing literature in neurotypical populations [25]. Carefully calibrated dosing will be important in future clinical trials, particularly for autistic individuals who may have different sensitivities [19]. Further, the meaningfulness of the psychedelic experience emerged as critical in determining the duration of mental health improvements in autistic adults. Our participants who rated their experiences as highly meaningful were more likely to report lasting benefits, emphasizing that the subjective quality of the experience is central to its therapeutic process, similar to those found in neurotypical adults [42]. The associations between dosage, duration of effects, and the meaningfulness of experiences highlight the need for well-designed studies to explore optimal dosing strategies and the qualitative aspects of psychedelic experiences in autistic people.

Interestingly, while there was a significant relationship between the type of psychedelics used and the duration of mental health improvements, we did not find a significant association between the type of psychedelics and the perceived personal or spiritual significance of the experience in autistic adults. This suggests that in autistic adults, the therapeutic benefits may be more influenced by the mindset and context (i.e., "set and setting" [43]) of the user than by the specific pharmacological properties of the drug used, a view that aligns with current research in neurotypical populations [19]. For autistic individuals, in particular, tailored settings that cater to their sensory and cognitive profiles may maximize therapeutic potential [13,32].

A key limitation of this study is potential sampling bias. Because participants were recruited primarily through online communities and self-selected into a survey about psychedelics, the sample may overrepresent individuals with prior interest, knowledge, or positive attitudes toward psychedelic use. This bias is further suggested by the recruitment strategy, which included psychedelic community networks, likely contributing to the high rate of prior psychedelic use observed. Moreover, given that 70% of respondents reported prior psychedelic use, these findings may not generalize to all autistic adults, particularly those who are psychedelic-naïve or less engaged in online mental health networks. In addition, the study's findings are limited by the racial homogeneity of the sample (87% White participants) and its focus on North America (87.7% of participants residing in Canada [45.6%] or the USA). This lack of diversity is consistent with broader underrepresentation of racial minorities in psychedelic research [45] and underscores the need for future studies to recruit participants from a wider range of racial and ethnic backgrounds, enhancing the generalizability and equity of findings [13]. Future research should employ targeted outreach strategies, such as partnering with organizations serving Black, Indigenous, and People of Color (BIPOC) communities and providing translated materials, to ensure more diverse representation. Interestingly, our Canadian participants differed from non-Canadians in perceived knowledge, willingness to try psychedelics, perceived helpfulness, prior use of multiple substances, and uncertainty about local decriminalization status—differences that may result from sampling bias or specific patterns among autistic Canadians.

Furthermore, our sample likely overrepresents autistic adults with higher education, English fluency, and fewer support needs, as participation required online access, English proficiency, and detailed self-reporting. We did not collect data on formal autism support needs levels (e.g., DSM-5 Levels 1, 2, or 3). This limits the scope and transferability of the findings, as individuals with greater support needs may hold different perspectives or face distinct risks when considering psychedelic use [33]. Future studies should prioritize accessible study designs, including non-text-based methods or facilitated interviews, to include individuals with communication differences or greater support needs. Social media-based recruitment may have further contributed to these biases by oversampling particular subgroups on the spectrum [48]. Furthermore, the study relied entirely on retrospective self-report for subjective experiences, dosages, and mental health outcomes. These reports are subject to recall bias and inaccuracies, particularly for experiences that occurred years earlier, and lack objective clinical validation. Some survey items, particularly those asking which mental-health conditions improved after psychedelic use, may

have been positively framed. These question formats could interact with participants' pre-existing expectations or beliefs about psychedelics, potentially influencing how they evaluated their experiences. Future surveys should balance question framing and include neutral or negatively framed items to mitigate expectation effects.

Safety considerations are particularly important when interpreting these results. Many participants reported using psychedelics outside of clinical or research settings, which may pose additional risks for autistic individuals, including heightened sensory sensitivities, anxiety, or co-occurring mental health challenges. These risks may be amplified in individuals with greater support needs. These findings emphasize the need for future psychedelic research and therapeutic applications to include clinical supervision, appropriate screening, and integration support tailored to autistic participants' needs.

Another key limitation is that our survey did not assess changes in autism-specific traits, such as social communication or restricted interests. The focus was intentionally placed on co-occurring mental health conditions rather than autism itself, consistent with our study's scope and ethical constraints. As a result, the data cannot speak to whether psychedelics directly affect autism-related characteristics. Nevertheless, one of the most notable findings was that 63.4% of participants reported improved relationships following psychedelic experiences, suggesting potential benefits for social connectedness that warrant further study. Lastly, because we did not collect data on the timing of adverse effects—opting to keep the survey brief and accessible—any immediate-versus-delayed outcomes remain unknown. Together, these considerations, along with self-selection bias, lack of a control group, and reliance on self-report data, underscore the need for caution when interpreting our results.

In conclusion, this study provides critical insights into autistic adults' interest in and experiences with psychedelics, suggesting strong curiosity and perceived potential for mental health support. The alignment of our findings with those of [33] reinforces the potential of psychedelics as a therapeutic tool for autistic adults. Our findings suggest that while dosage and the meaningfulness of the experience play crucial roles in determining therapeutic outcomes in autistic adults, significant barriers (including legal status, perceived health risks, and logistical concerns) to participation remain. By addressing identified barriers and enhancing education and familiarity with psychedelics, future research grounded in lived experience can pave the way for more inclusive and effective approaches. This will help ensure that all diverse communities, particularly the frequently marginalized autistic population, can have equitable access to potential improvements in well-being through psychedelic therapies, informing future public mental health policy [13,45]. These findings highlight the importance of continued exploration into the unique needs of autistic individuals in the therapeutic application of psychedelics.

## Supporting information

**S1 Text. Full text of the online survey administered to participants.**
(DOCX)

**S1 Table. Knowledge and perception, stratified by sex, education, age, marital status, and country of residence.**
(DOCX)

**S2 Table. General psychedelic experiences, stratified by sex, education, age, marital status, and country of residence.**
(DOCX)

**S3 Table. Psychedelic experience that led to improvements in your mental health, stratified by sex, education, age, marital status, and country of residence.**
(DOCX)

**S4 Table. Feelings, thoughts, and experiences during the overall psychedelic experience that led to improvements in your mental health, stratified by sex, education, age, marital status, and country of residence.**
(DOCX)

## Acknowledgments

We gratefully acknowledge the participants who shared their time and experiences, as well as the autistic advisors at Azrieli Adult Neurodevelopmental Centre at CAMH.

## Author contributions

**Conceptualization:** Ami Tint, Hsiang-Yuan Lin.

**Data curation:** Sahba Afsharnia, Vivian Liang.

**Formal analysis:** Vivian Liang.

**Funding acquisition:** Ami Tint, Hsiang-Yuan Lin.

**Investigation:** Sahba Afsharnia, Hsiang-Yuan Lin.

**Methodology:** Sahba Afsharnia, Ami Tint, Hsiang-Yuan Lin.

**Project administration:** Hsiang-Yuan Lin.

**Resources:** Yona Lunsky, Aaron P. Orsini, Ami Tint.

**Software:** Sahba Afsharnia.

**Supervision:** Hsiang-Yuan Lin.

**Validation:** Hsiang-Yuan Lin.

**Visualization:** Vivian Liang.

**Writing – original draft:** Sahba Afsharnia.

**Writing – review & editing:** Vivian Liang, Yona Lunsky, Aaron P. Orsini, Ami Tint, Hsiang-Yuan Lin.

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
