## [Decision Letter · Decision Letter 0]

15 Aug 2025

PMEN-D-25-00172

Knowledge, Perceptions, and Use of Psychedelics among autistic adults: An online survey

PLOS Mental Health

Dear Dr. Lin,

Thank you for submitting your manuscript to PLOS Mental Health. After careful consideration, we feel that it has merit but does not fully meet PLOS Mental Health’s publication criteria as it currently stands. Therefore, we invite you to submit a revised version of the manuscript that addresses the points raised during the review process.

We look forward to receiving your revised manuscript.

Kind regards,

Joel Frohlich, PhD

Academic Editor

PLOS Mental Health

Journal Requirements:

i. State the initials, alongside each funding source, of each author to receive each grant.

2. Please ensure that your Ethics Statement is available in its entirety at the beginning of your Methods section, under a subheading 'Ethics Statement'.

3. Tables should not be uploaded as individual files. Please remove these files and include the Tables in your manuscript file as editable, cell-based objects. For more information about how to format tables, see our guidelines: 

https://journals.plos.org/mentalhealth/s/tables

4. We have noticed that you have uploaded Supporting Information files, but you have not included a list of legends. Please add a full list of legends for your Supporting Information files after the references list.

Additional Editor Comments (if provided):

Reviewers' comments:

Reviewer's Responses to Questions

**Comments to the Author**

1. Does this manuscript meet PLOS Mental Health’s publication criteria?

Reviewer #1: Yes

Reviewer #2: Yes

Reviewer #3: Partly

Reviewer #4: Partly

2. Has the statistical analysis been performed appropriately and rigorously?

Reviewer #1: Yes

Reviewer #2: Yes

Reviewer #3: Yes

Reviewer #4: N/A

3. Have the authors made all data underlying the findings in their manuscript fully available (please refer to the Data Availability Statement at the start of the manuscript PDF file)?

Reviewer #1: Yes

Reviewer #2: Yes

Reviewer #3: Yes

Reviewer #4: No

4. Is the manuscript presented in an intelligible fashion and written in standard English?

Reviewer #1: Yes

Reviewer #2: Yes

Reviewer #3: Yes

Reviewer #4: Yes

Reviewer #1: This manuscript presents a comprehensive cross-sectional survey exploring autistic adults’ knowledge, perceptions, and experiences with psychedelics. It addresses a significant gap in the literature by focusing on a marginalized and underrepresented group in clinical psychedelic research. The work is grounded in a strong rationale, methodologically rigorous, and offers novel insights relevant to both clinical research and public health.

see the attached for more details

Reviewer #2: In general this was a well-written and designed paper focussed on an interesting and novel area. I only have comments regarding minor changes.

1. Data Availability. Authors link an OSF repository and state the data from the study is available here: https://osf.io/zceyk/ however on accessing the link there is no data to download. I have still selected "Yes" in answer to question 3 above because the authors also include supplementary tables with survey responses in word format. Please remedy the OSF link issue.

2. Abstract. Related to comments below regarding the specificity of this sample and the generalisability of the conclusions, I recommend specifying the conclusions may not apply to all autistic adults e.g., “Autistic adults *in this self-selecting sample* demonstrated strong interest...". (lines 37-38, p. 2).

3. Introduction. Very clear and generally well-written. Only minor recommendations:

a. "Ayahuasca" should not be capitalised (line 62, p. 3).

b. Rephrase for English: “Only a pilot trial in modern-era research explored the use of MDMA in a small sample of autistic adults (N = 12; 8 in the MDMA group), demonstrating a promising signal of MDMA’s reducing social anxiety and good safety and tolerability (32).” (lines 78-80, p. 3). I know what you mean but needs to be rephrased.

c. I recommend including reference to negative side effects from these early studies for balance. Significant negative effects were also reported. See Markopoulos (2022), Bender et al. (1966) – e.g. aggression, biting, self-harm etc.

d. “However, outcomes varied significantly based on individual mindsets, knowledge, and expectations.” This statement is not supported by the referenced study. (lines 86-87, p. 4). Either change the statement or provide a reference that supports it.

e. “Also, whether this first survey results are reliable in different autistic cohorts remains unknown.” Rephrase for English. E.g. “the results of this first survey” reads better than “this first survey results”. (lines 90-91, p. 4)

f. “(1) What are the general perspectives and knowledge levels of autistic adults on psychedelics?” – rephrase for English. (lines 94-95 p.4), “knowledge levels of autistic adults on psychedelics” does not read well.

g. In your second research question please clarify which type of symptoms you are looking for opinions on – autistic symptoms? Mental health condition symptoms? Both?

4. Materials and Methods. Great and clear section. Only minor recommendations:

a. Remove additional ‘were’ from this phrasing “2) self-identified or were professionally diagnosed with ASC,” (lines 102-103, p. 4)

b. Specify which social media sites and community organisations you recruited from to better describe the sample. This can be very meaningful in online surveys. For example were the participants recruited from autism advocacy organisations? Were they recruited from psychedelics advocacy organisations or subreddits? This is an important descriptor for potential bias in the sample. Especially if in the discussion or abstract it is implied these findings may apply to the autistic population as a whole.

5. Results. Clear and well-written. Only one recommendation: for concision and lack of repetition of data in the tables you could cut some of the main text and reserve it for highlighting only the most significant findings in the tables.

6. Discussion. Very clear, well-written and interesting. Only minor recommendations:

a. Most importantly, this paper needs more discussion around the possibility (probability) that the sample is significantly biased. I recommend this is also highlighted in the Abstract. For example, this was not a psychedelic naive sample (70% having used a psychedelic previously). Moreover, purely by the nature of the topic of survey (psychedelics) respondents may be more likely to answer if they already have an interest in the area. These findings may not generalise to the wider autistic population and this needs to be highlighted in the paper as a key limitation.

b. Relatedly, the following statement, "In terms of public perceptions, while the neurotypical population may harbor legal and safety concerns (37), our participants showed more willingness to engage in research, suggesting that autistic adults may be more open to alternative mental health treatments." (lines 361-363, p. 10) seems to imply that autism explains greater openness to alternative MH treatments, whilst it may in fact be due to the fact you have sampled a group of pro-psychedelic participants. Either modify the statement or provide evidence that it is in fact autism and not other factors (e.g. previous psychedelic use or recruitment from psychedelic related websites or organisations) causing this.

c. Discussion of the limitations of self-reported dosage in online surveys is also needed in addition to the limitations of your chosen assessment of dose. Please make it clear in the methods that dosage was assessed in a very low resolution fashion and provide a short justification for this method of dose assessment.

General comments: An interesting paper using sound methodology and statistics investigating a novel area. It would be improved by including greater discussion and highlighting of the above. Thank you for your work on this.

Reviewer #3: Overall Assessment:

This study addresses an important and understudied area in psychedelic research by examining the knowledge, perceptions, and experiences of autistic adults with psychedelics. The work fills a significant gap in our understanding of how this population might respond to and benefit from psychedelic therapies. While the research provides valuable insights into autistic adults' perspectives on psychedelics, several methodological concerns limit the generalizability and reliability of the findings.

Strengths:

The authors have undertaken a valuable investigation into a population that has been historically excluded from clinical trials despite facing high rates of co-occurring mental health conditions. The large sample size of completed surveys provides substantial data on autistic adults' experiences with psychedelics. The co-production approach involving autistic advisors in survey development demonstrates appropriate community engagement and enhances the accessibility of the research instrument. The finding that a majority of participants had used psychedelics, with the majority reporting mental health benefits, provides important preliminary evidence about real-world usage patterns in this population.

Major Concerns:

The recruitment strategy and self-report paradigm likely introduced significant selection bias toward higher-functioning autistic individuals. Online surveys requiring English fluency and the ability to provide detailed self-reports naturally exclude individuals with greater support needs. The predominantly female sample likely interacts with autism support levels in complex ways that remain unaddressed in the analysis. Most critically, the study fails to account for autism support needs levels (formerly functioning levels 1, 2, and 3), which is a significant omission given that individuals requiring substantial or very substantial support may face different risks when using psychedelics without clinical supervision.

The retrospective nature of the data collection raises significant concerns about recall bias, as participants may have difficulty accurately remembering experiences that occurred years ago. The study relies entirely on subjective self-report without objective validation or clinical assessment, which limits the reliability of reported mental health improvements and adverse effects.

Several survey questions appear potentially leading in their framing toward improvement, such as "Which of these mental health conditions improved after using psychedelics?" and "Did your past psychedelic use help with your mental health concerns?" While this approach is understandable given the study's focus, the limitations of this framing should be more explicitly acknowledged. Also, the temporal direction of expectation effects remains unclear - participants may have had pre-existing expectations about psychedelics that influenced their experiences, or they may have learned about potential benefits after use, which could color their retrospective reporting.

Safety Considerations:

While the authors discuss barriers to psychedelic use, they inadequately address the potential risks of unsupervised psychedelic use among autistic individuals, particularly those with greater support needs. The discussion should more explicitly outline safety concerns and emphasize the importance of clinical supervision, especially given the heightened vulnerabilities in this population including sensory sensitivities and co-occurring mental health challenges.

Reviewer #4: This is a well-written manuscript about survey responses related to classical psychedelic and MDMA use for mental health conditions that may co-occur with autism. The survey was administered to individuals who may have identified as being autistic. However, the actual percentage of individuals who endorsed being autistic was not presented in the results or any tables, which is concerning. Aside from this one question asking whether the respondent identified as autistic, there was only one other question related to autism, which was how helpful respondents thought psychedelics and MDMA could be in treating depression, anxiety, or PTSD or other mental health conditions (which aren’t autism), but specifically for people with autism. Regardless if all or none of the respondents actually identified as autistic, in light of the limited scope of the design of the survey, the manuscript fails in three key areas, 1) in terms of its three primary research questions posed at the end of the introduction, 2) in terms of its central aim “to explore knowledge, perceptions, and experiences of autistic adults regarding psychedelics” and 3) in terms of is title, “Knowledge, Perceptions, and Use of Psychedelics among autistic adults: An online survey.” Since it may be too late to develop additional questions that would satisfy these three key areas adequately, the only way forward is to reframe the manuscript and revise the language in these three areas (edit the title, aims, and research questions), to accurately reflect the work that was done so that readers are not disappointed by what they will not find.

1. Abstract – Define “neurotypical” at first use, as mental health conditions are forms of neurodivergence and affect neurodevelopmental trajectories, which may confuse readers when insinuating that individuals with depression, etc., are neurotypical.

2. Abstract – Be explicit that although the therapeutic potential for treating autism remains under-explored, this work was designed to merely address the co-occurring health conditions in autism and not autism itself.

3. Abstract – The abstract would benefit from a brief description of how “psychedelics” were defined in the survey. Some readers may be curious to know whether MDMA was included, for instance, since there has been some research with MDMA in this population.

4. Introduction – This is what I wrote before I discovered the limited scope of the survey: “The first paragraph reads as if the objective of the study is to determine how individuals with autism might merely treat their co-occurring mental health conditions with psychedelics. There is no mention of treating the symptoms of autism itself, how autism is currently treated, gaps in knowledge with treatments, etc., to lead up to potential for psychedelics, including MDMA, as a promising treatment for autism. It isn’t until the final paragraph of the introduction, when the three primary research questions are introduced, does the reader feel oriented, and relieved, that “opinions on using psychedelics as a treatment for their symptoms” indeed falls within the scope of the study objectives.” This original comment shows what readers think the manuscript was aiming to do, when in reality, it was true that the objective was merely to determine how autistic individuals treat co-occurring mental health conditions. Therefore, instead of following my original instructions prior to learning about the limited scope, please be more explicit that the objective is not about autism itself. This will be less problematic after the title, aims, and research questions are revised.

5. Introduction – For greater accuracy, ayahuasca is not a classical psychedelic, but a brew containing a classical psychedelic (DMT). It also should not be capitalized.

6. Methods – “CAMH” is not defined at first mention but is defined at the end of the methods. Perhaps, as a solution to this small issue, that final section in the methods could be brought up and combined with the Participants section so the reader also has the information related to ethical approvals immediately with the participants, which is more standard.

7. Results – A reminder to include the percent of respondents who identified as autistic. Were only the responses from this percent used in the rest of the manuscript? Or were all responses used, regardless if the respondent identified as autistic? If the latter, please restructure findings to focus only on those who responded as identifying as autistic and include not only the number of completers of the survey, but also the number of individuals who identified as autistic as the final number of respondents analyzed.

8. Results – This sentence is missing “co-occurring” before mental health conditions: “A significant portion of the sample reported having a history of mental health conditions (96.6%)”

9. Results – I’m not sure I understand this sentence, “…who initially endorsed persisting negative effects, four did not describe substantive psychedelic-related issues (e.g., “Good,” “Leading questions,” “Initially depressed but became better,” “Hallucination from 1980s marijuana”).” I think the “e.g.” is meant to be an “i.e.” here, to show what the four descriptions were. I also think these four descriptions could be removed as they seem to do more to confuse than add to the understanding of the sentence. Or just keep “initially depressed but became better” after “e.g.,” with the other three removed. Moreover, it’s also unclear what is meant by “initially endorsed” here. If an earlier item also asked about persisting negative effects for some reason, this should be made explicit.

10. Discussion – The discussion must address the key limitation of the survey, that no questions were asked about how psychedelics or MDMA affect autism itself. Moreover, this limitation should reveal that only one question related to autism was posed in the survey and describe what that question asked. These limitations may be offset by what I see as the most powerful and interesting finding in the paper, which was that “The most frequently reported change was improved relationships, as noted by 63.4% of participants,” this is the greatest contribution of the work and especially relevant for autism/ASC.

11. Discussion – Instead of the word “trip” use “experience”

**Do you want your identity to be public for this peer review?** For information about this choice, including consent withdrawal, please see our Privacy Policy

Reviewer #1: **Yes: ** Dr David Onchonga

Reviewer #2: No

Reviewer #3: No

Reviewer #4: No

---

## [Decision Letter · Decision Letter 1]

26 Nov 2025

Knowledge, Perceptions, and Use of Psychedelics for Mental Health among Autistic Adults: An Online Survey

PMEN-D-25-00172R1

Dear Dr. Lin,

We are pleased to inform you that your manuscript 'Knowledge, Perceptions, and Use of Psychedelics for Mental Health among Autistic Adults: An Online Survey' has been provisionally accepted for publication in PLOS Mental Health.

Best regards,

Joel Frohlich, PhD

Academic Editor

PLOS Mental Health

Reviewer Comments (if any, and for reference):

Reviewer's Responses to Questions

**Comments to the Author**

Reviewer #3: All comments have been addressed

Reviewer #4: All comments have been addressed

publication criteria?

Reviewer #3: Yes

Reviewer #4: Yes

3. Has the statistical analysis been performed appropriately and rigorously?

Reviewer #3: Yes

Reviewer #4: Yes

4. Have the authors made all data underlying the findings in their manuscript fully available (please refer to the Data Availability Statement at the start of the manuscript PDF file)?

Reviewer #3: Yes

Reviewer #4: Yes

5. Is the manuscript presented in an intelligible fashion and written in standard English?

Reviewer #3: Yes

Reviewer #4: Yes

Reviewer #3: The authors have successfully addressed all of my concerns.

Reviewer #4: No further comments.

**Do you want your identity to be public for this peer review?** For information about this choice, including consent withdrawal, please see our Privacy Policy

Reviewer #3: No

Reviewer #4: No
